# The order and logic of CD4 versus CD8 lineage choice and differentiation in mouse thymus

Mohammad M. Karimi [1,5], Ya Guo[1,6], Xiaokai Cui[1], Husayn A. Pallikonda [1], Veronika Horková [2], Yi-Fang Wang[1], Sara Ruiz Gil[3], Gustavo Rodriguez-Esteban[3], Irene Robles-Rebollo[1], Ludovica Bruno[1], Radina Georgieva[1], Bhavik Patel[1], James Elliott[1], Marian H. Dore[1], Danielle Dauphars[4], Michael S. Krangel[4], Boris Lenhard [1], Holger Heyn [3], Amanda G. Fisher [1], Ondřej Štěpánek [2] & Matthias Merkenschlager [1] ✉

CD4 and CD8 mark helper and cytotoxic T cell lineages, respectively, and serve as coreceptors for MHC-restricted TCR recognition. How coreceptor expression is matched with TCR specificity is central to understanding CD4/CD8 lineage choice, but visualising coreceptor gene activity in individual selection intermediates has been technically challenging. It therefore remains unclear whether the sequence of coreceptor gene expression in selection intermediates follows a stereotypic pattern, or is responsive to signaling. Here we use single cell RNA sequencing (scRNA-seq) to classify mouse thymocyte selection intermediates by coreceptor gene expression. In the unperturbed thymus, *Cd4+Cd8a-* selection intermediates appear before *Cd4-Cd8a+* selection intermediates, but the timing of these subsets is flexible according to the strength of TCR signals. Our data show that selection intermediates discriminate MHC class prior to the loss of coreceptor expression and suggest a model where signal strength informs the timing of coreceptor gene activity and ultimately CD4/CD8 lineage choice.

[1] MRC London Institute of Medical Sciences, Institute of Clinical Sciences, Faculty of Medicine, Imperial College London, London, UK. [2] Laboratory of Adaptive Immunity, Institute of Molecular Genetics of the Czech Academy of Sciences, Prague, Czech Republic. [3] CNAG-CRG, Centre for Genomic Regulation (CRG), The Barcelona Institute of Science and Technology (BIST), Barcelona, Spain. [4] Department of Immunology, Duke University Medical Center, Durham, NC, USA. [5] Present address: Comprehensive Cancer Centre, School of Cancer & Pharmaceutical Sciences, Faculty of Life Sciences & Medicine, King's College London, London, UK. [6] Present address: School of Life Sciences and Biotechnology, Shanghai Jiao Tong University, Shanghai, China. ✉email: matthias.merkenschlager@lms.mrc.ac.uk

C D4 helper T cells and CD8 cytotoxic T cells are the two principal T cell lineages in the mammalian immune system. Although distinct in phenotype and function, CD4 and CD8 T cells arise in the thymus from bi-potential, $CD4^+$ $CD8^+$ double-positive (DP) progenitors. CD4/CD8 lineage choice is critical for the generation and selection of the T cell receptor (TCR) repertoire, and represents one of the most intensely studied and enigmatic examples of a binary lineage decision. TCR rearrangement is fundamentally stochastic, and the functional properties of the newly generated TCR repertoire are established empirically during positive and negative selection. CD4 and CD8 act as coreceptors for TCR recognition restricted by major histocompatibility complex (MHC) class II or class I, respectively. Competing models have been proposed for how coreceptor expression is matched with TCR specificity, and differ with respect to the mode of lineage branching and the role of TCR signal strength in MHC class discrimination. One class of models envisage symmetric branching of the CD4 and CD8 lineages where lineage choice is either instructive[1] or stochastic[2,3]. Alternatively, kinetic signalling models posit that CD4/CD4 lineage choice occurs in a series of sequential steps whereby TCR-signalled thymocytes initially downregulate *Cd8* to audition for the CD4 lineage, and switch coreceptor expression from $Cd4^+Cd8^-$ to $Cd4^-Cd8^+$ only if they experience a loss of CD8-dependent TCR signalling[4–8]. MHC class discrimination by signal strength is critical in instructive models[1], but entirely dispensable in stochastic/selective models of CD4/CD8 lineage choice[2,3]. Proposals of CD4/CD8 lineage choice driven by signal strength and duration[9–14] have been largely superseded by the idea that MHC class discrimination is based solely on signal continuity during sequential expression of coreceptors[4,6,7,15,16].

Lineage commitment occurs at the level of individual progenitor cells, and is therefore inherently difficult to study at the population level. The expression of *Cd4* and *Cd8* coreceptor genes by individual selection intermediates is central to understanding CD4/CD8 lineage choice, but has never been observed directly. Previous single-cell RNA-seq studies of the thymus have largely ignored cells in transition from bipotential progenitors to the CD4 SP or the CD8 SP stages[17–19], in part because such selection intermediates are rare and make up only a few percent of cells in the thymus. High throughput scRNA-seq approaches may profile gene expression by large numbers of cells lack often the depth required to unambiguously assign *Cd4* and/or *Cd8* expression to individual selection intermediates, while scRNA-seq approaches that deliver sufficient depth have limited throughput.

Here we address this challenge by prospective isolation of selection intermediates combined with scRNA-seq of full length transcripts. This approach provides a direct view of *Cd4* and *Cd8a* coreceptor gene activity in individual selection intermediates within a framework of maturation, activation and lineage specification during CD4/CD8 lineage choice. The resulting data are broadly in line with models of sequential lineage determination by kinetic signalling, albeit with substantial refinements and modifications. In the unperturbed thymus, the order of coreceptor gene expression states is $Cd4^+$ $Cd8a^+$ followed by $Cd4^+$ $Cd8a^-$ and later by $Cd4^-$ $Cd8a^+$, providing direct support for sequential coreceptor gene expression during CD4/CD8 lineage choice. Interestingly, however, perturbation experiments reveal that selection intermediates discriminate between MHC classes prior to the loss of *Cd4* or *Cd8* coreceptor expression, and accelerate their transition to the $Cd4^-Cd8a^+$ state in response to weaker TCR signals in the absence of MHC class II. These findings suggest a model that links signal strength and the timing of coreceptor gene expression.

## Results

**scRNA-seq of CD4 CD8 lineage choice and differentiation.** We performed scRNA-seq of normal thymocytes from unmanipulated adult mice at steady-state. To address patterns of coreceptor gene activity and associated gene expression programs, we opted for deep sequencing of full-length transcripts by SMART-seq. This approach identifies thousands of transcripts per cell at sufficient depth to reliably call coreceptor expression by individual selection intermediates (see below). To capture sufficient numbers of selection intermediates, we isolated single cells representing thymocytes before ($CD69^-$ DP), during ($CD69^+$ DP, $TCR\beta^{hi}$ DP, $CD4^+$ $CD8^{low}$), and after lineage choice and differentiation (CD4 SP, $TCR\beta^{hi}$ CD8 SP) from wild-type C57BL/6 mice (Supplementary Fig. 1). Full-length single-cell RNA-seq libraries were prepared and sequenced, identifying thousands of transcripts per cell (Supplementary Table 1, see Methods). We identified highly variable genes (Supplementary Data 1) and removed *Cd4, Cd8a/b1* for later use in the classification of selection intermediates (see below). Principal component analysis (PCA) showed that PC1, PC2 and PC3 accounted for 44.66%, 21.60% and 13.79% of differential gene expression among the first five PCs (Fig. 1a). The first principal component, PC1, positioned sorted thymocyte subsets in order of maturity: pre-selection thymocytes ($CD69^-$ DP, red) on the left, CD4 SP (green) and CD8 SP (blue) on the right. Selection intermediates ($CD69^+$ DP, orange, $TCR\beta^{hi}$ DP, grey, and $CD4^+$ $CD8low$, yellow) were in the centre of PC1 (Fig. 1a). The observed patterns were highly reproducible between replicates (Supplementary Fig. 2).

**scRNA-seq captures maturation, activation and lineage identity as the first three principal components of CD4/CD8 lineage choice and differentiation.** To validate the approach, we examined differentially expressed genes with well-defined expression patterns during CD4/CD8 lineage choice and differentiation (Supplementary Fig. 3a and Supplementary Data 2). *Rag1* and *Dntt* are active in pre-selection DPs where they contribute to the somatic rearrangement of the *Tcra/d* locus. scRNA-seq found highly correlated *Rag1* and *Dntt* expression in individual cells ($R = 0.75$, $P < 2.2e-16$; Supplementary Fig. 3b). The expression of genes that are highly active in pre-selection DP thymocytes (e.g. *Rag1, Dntt, Cd24a*) progressively decreased along PC1 (Fig. 1b). The expression of genes associated with thymocyte maturation such as *Ccr7, Il7r B2m* and the MHC class I genes *H2-D1* and *H2-K1* progressively increased along PC1 (Fig. 1c). *Rag1* and *Dntt* are silenced by TCR signals, and their expression is superseded by activation markers such as *Cd69*. *Rag1* and *Cd69* were anti-correlated in individual cells by scRNA-seq ($R = -0.27$, $P = 1.7e-09$, Supplementary Fig. 3c). Maturation markers such as *Ccr7* and *Il7r* were positively correlated in individual cells ($R = 0.38$, $P < 2.2e-16$, Supplementary Fig. 3d). Genes that showed non-linear behaviour during thymocyte differentiation featured strongly in the second principal component, PC2, which reflects the transient induction and repression of genes by TCR signalling such as *Cd69, Cd5, Tox, Cd28* and *Itm2a* (Fig. 1c). *Cd4* and *Cd8a* were co-expressed in pre-selection DP but mutually exclusive in CD4 and CD8 SP (Supplementary Fig. 3e). *Cd4, Cd8a* and other markers of the CD4 and CD8 lineages segregated in the third principal component PC3, which showed a branched trajectory from the pre-selection DP stage ($CD69^-$ DP) through intermediate $CD69^+$ DP and $CD4^+$ $CD8^{low}$ stages towards CD4 and CD8 single-positive populations (Fig. 1d). Although maturation, activation and lineage identity are recognised as central components of lineage choice and differentiation in the thymus, our analysis quantifies the relative contribution of each (maturation, activation and lineage identity) to

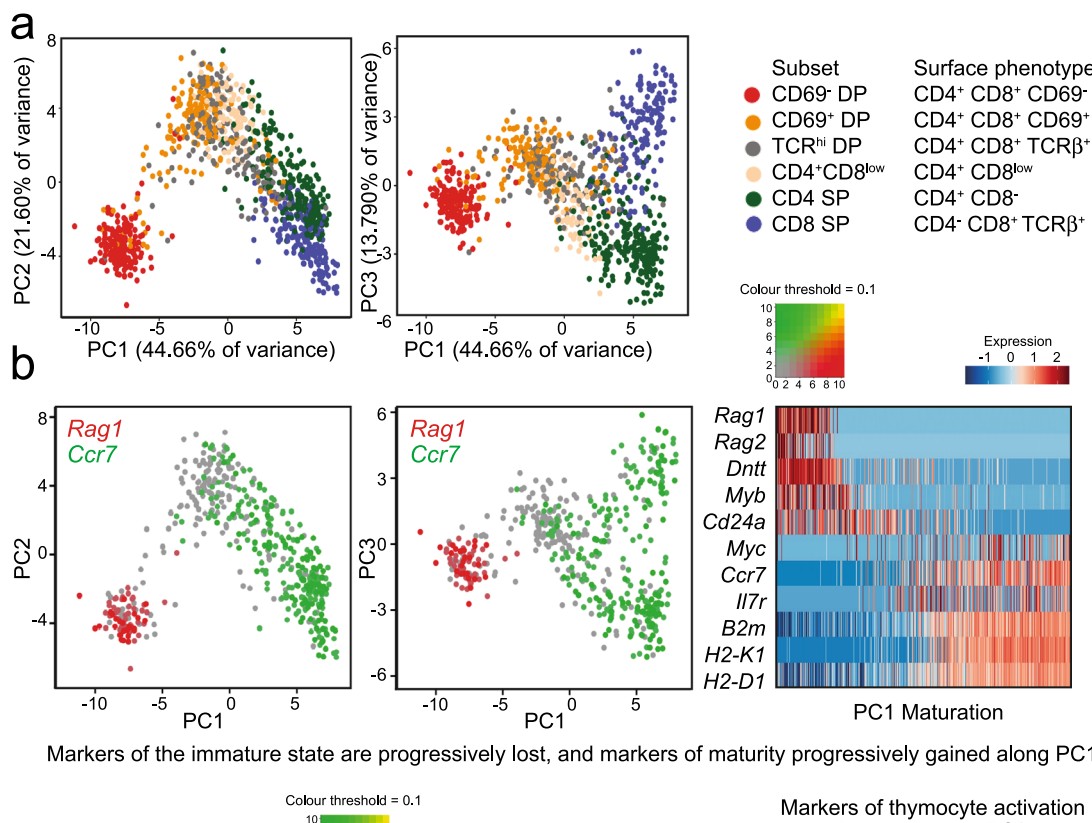

Markers of the immature state are progressively lost, and markers of maturity progressively gained along PC1

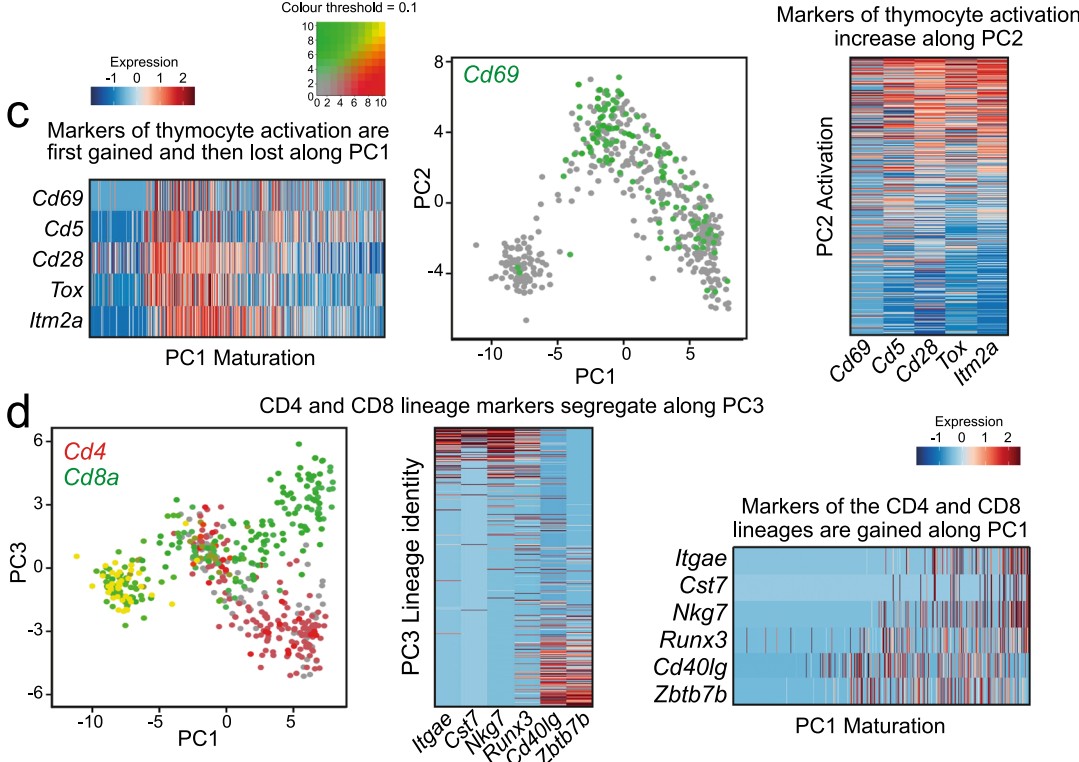

the overall variance of gene expression, and shows that scRNA-seq data alone are sufficient to derive a coherent framework for CD4/CD8 lineage choice and differentiation.

To challenge the interpretation of PC1 as maturation, PC2 as activation and PC3 as CD4/CD8 lineage identity, we identified genes associated with thymocyte maturation, activation and CD4/CD8 lineage choice derived from population RNA-seq for use as an external reference. We defined activation genes as genes that

were transiently up- or downregulated between the CD69⁻ DP stage and the CD4 and CD8 SP stages of differentiation ($P < 0.05$, $n = 524$), and CD4 and CD8 lineage genes as genes that were differentially expressed between CD69⁻ CD4SP and TCRʰⁱ CD8SP ($P < 0.05$, $n = 344$; Supplementary Data 3). We then calculated the enrichment of activation genes and lineage genes. PC1 was not enriched for activation ($P = 0.327$, Fisher Exact test, top 1000 highly correlated genes with PC1) and moderately

**Fig. 1 scRNA-seq captures maturation, activation and lineage identity as principal components of CD4/CD8 lineage choice and differentiation.**
**a** Highly variable genes in the scRNA-seq data were identified and principal component analysis (PCA) was performed, excluding *Cd4, Cd8a* and *Cd8b1*. PC1 versus PC2 (left) and PC1 versus PC3 (right) are shown, and the percentage of variance explained by each PC is indicated as a percentage of the first five PCs. PC4 and PC5 explained 11.45% and 8.50% of variance within the first five PCs and 5.82% and 4.32% of total variance, respectively. Sorted subsets are coloured as indicated in the key. Cell numbers are shown in Supplementary Table 2. Source data are provided as a Source Data file. **b** PC1 reflects progressive thymocyte maturation. A two-colour dot plot projected onto a map of PC1 versus PC2 shows expression of *Rag1*(red), which was confined to pre-selection thymocytes, and *Ccr7*, which was acquired during maturation. A heat map of key thymocyte maturation genes (right) shows how markers of immature thymocytes are progressively lost along PC1 and markers of mature thymocytes are progressively gained along PC1. Source data are provided as a Source Data file. **c** PC2 reflects transient thymocyte activation. Expression of the activation marker *Cd69* (green) projected onto a map of PC1 versus PC2. Heat maps of key thymocyte activation genes (right) show how markers of thymocyte activation are progressively gained along PC2. The same markers show non-linear behaviour along PC1, as they are first gained and then lost during thymocyte maturation. Source data are provided as a Source Data file. **d** PC3 reflects CD4/CD8 lineage identity. Expression of *Cd4* (red) and *Cd8a* (green), as a two-colour dot plot projected onto a map of PC1 versus PC3. A heat map of CD4- and CD8 lineage-specific genes (right) shows how PC3 segregates markers of the CD4 and CD8 lineages. Source data are provided as a Source Data file.

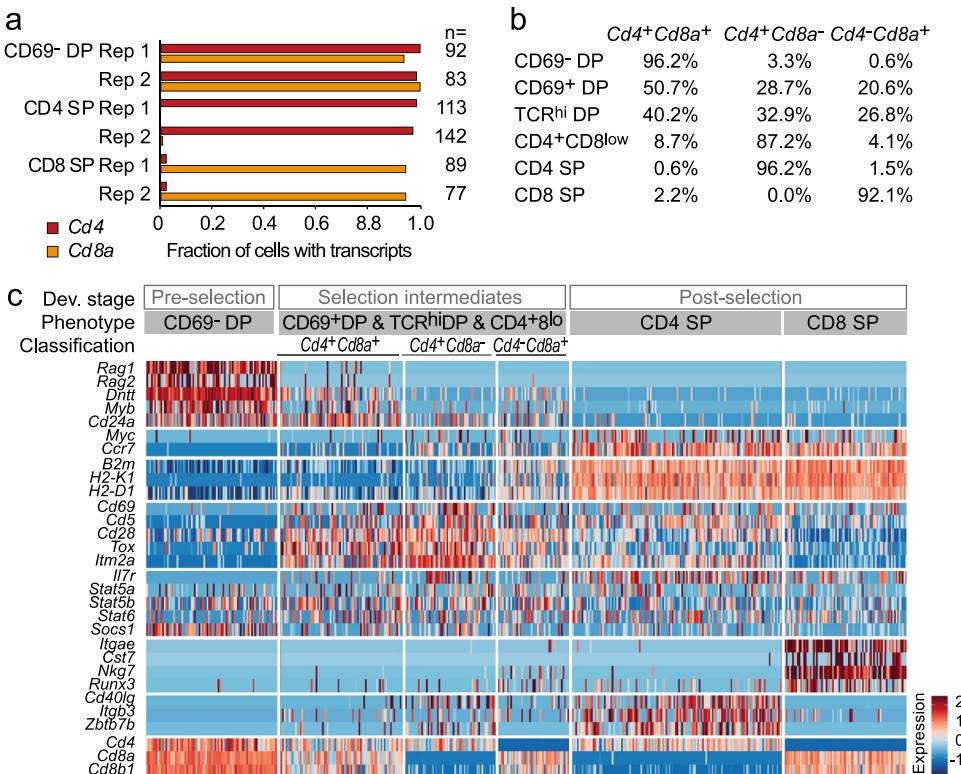

**Fig. 2 scRNA-seq reliably detects the coreceptor transcripts *Cd4* and *Cd8a*. a** scRNA-seq detection frequencies of *Cd4* (red) and *Cd8a* (orange) in sorted CD69⁻ DP, CD4 SP and CD8 SP wild-type thymocytes. Bars of the same colour represent independent biological replicates. Cell numbers are listed in Supplementary Table 2. Source data are provided as a Source Data file. **b** Representation of *Cd4⁺Cd8a⁺, Cd4⁺Cd8a⁻* and *Cd4⁻Cd8a⁺* coreceptor gene expression patterns in by CD69⁻ DP, CD69⁺DP, TCRβʰⁱ DP, CD4⁺CD8ˡᵒʷ, CD4 SP and CD8 SP wild-type thymocytes. **c** Cells (columns) are grouped by developmental stage (pre-selection, selection intermediates and post-selection) and cell surface phenotype. CD69⁻ DP represent pre-selection thymocytes, pooled CD69⁺DP, TCRβʰⁱ DP and CD4⁺CD8ˡᵒʷ represent selection intermediates, CD4 SP, TCRβʰⁱ CD8 SP represent post-selection thymocytes. Selection intermediates are classified into *Cd4⁺Cd8a⁺, Cd4⁺Cd8a⁻* and *Cd4⁻Cd8a⁺* based on scRNA-seq detection of *Cd4* and *Cd8a*. The data shown are for replicate 2. See Supplementary Table 7 for cell numbers.

enriched for CD4/CD8 lineage (*P* = 3.00e−06, Fisher Exact test, top 1000 highly correlated genes with PC1). PC2 was strongly enriched for activation (*P* = 2.22e−12, Fisher Exact test, top 1000 highly correlated genes with PC2), and only weakly for CD4/CD8 lineage (*P* = 0.0021, Fisher Exact test, top 1000 highly correlated genes with PC3). Finally, PC3 was strongly enriched for CD4/CD8 lineage (*P* = 1.15e−9, Fisher Exact test, top 1000 highly correlated genes with PC3), and only weakly for activation (*P* = 0.0055, Fisher Exact test, top 1000 highly correlated genes with PC3). This analysis validated that PC2 and PC3 reflect activation, and CD4/CD8 lineage identity, respectively.

**Classification of selection intermediates by *Cd4* and *Cd8a* coreceptor gene expression.** We next examined the ability of scRNA-seq to reliably detect coreceptor transcripts in pre- and post-selection thymocytes. scRNA-seq found expression of both *Cd4* and *Cd8a* (*Cd4* and *Cd8a*) in 93.5% (93.5% *Cd4⁺ Cd8a⁺*, replicate 1) and 98.8% (98.8% *Cd4⁺ Cd8a⁺*, replicate 2) of individual pre-selection DP thymocytes. scRNA-seq detected *Cd4* but not *Cd8a* in 98.6% (98.6% *Cd4⁺ Cd8a⁻*, replicate 1) and 96.2% (96.2% *Cd4⁺ Cd8a⁻*, replicate 2) of CD4 SP and *Cd8a* but not *Cd4* in 92.1% of CD8 SP (92.1% *Cd4⁻ Cd8a⁺*, replicates 1 and 2, Fig. 2a). These data indicate that scRNA-seq detects

appropriate constellations of coreceptor transcripts in the vast majority of pre- and post-selection thymocytes, and can therefore be used to classify selection intermediates into $Cd4^+ Cd8a^+$, $Cd4^+ Cd8a^-$, and $Cd4^- Cd8a^+$ (Fig. 2b). This allowed us to identify transcriptional programs of individual selection intermediates of defined coreceptor status (Fig. 2c, see Supplementary Data 4 for an analysis of differential gene expression between selection intermediates classified by coreceptor status and Supplementary Fig. 4 for a heatmap of gene expression by individual selection intermediates in MHC class II$^{-/-}$ thymus).

### Expression of TCR and cytokine signalling genes by selection intermediates of defined coreceptor expression status.

Kinetic signalling models predict that selection intermediates that terminate Cd8 expression react to a loss of CD8-dependent TCR signalling by increased responsiveness to cytokines, which results in the expression of Runx3 and the reversal of coreceptor gene expression[4]. In contrast to models where the differentiation of selection intermediates towards the CD8 lineage following CD4 to CD8 coreceptor reversal is exclusively driven by cytokine signals[20], the expression of TCR-driven activation genes such as Cd69, Egr1, Nr4a1 and Itm2a was significantly higher in $Cd4^+ Cd8a^+$, $Cd4^+ Cd8a^-$ and $Cd4^- Cd8a^+$ selection intermediates than in pre-selection DP (Fig. 3a). $Cd4^- Cd8a^+$ selection intermediates expressed slightly less Cd69, Egr1 and Itm2a, (but not Nr4a1) than $Cd4^+ Cd8a^-$ selection intermediates, but differences in TCR signalling gene expression between subsets of selection

intermediates were minor compared to the highly significant changes from pre-selection DP to selection intermediates (Fig. 3a). IPA pathway analysis indicated significant activation of pathways downstream of the TCR and of CD3 in both $Cd4^+ Cd8a^-$ and $Cd4^- Cd8a^+$ selection intermediates (Fig. 3b, left), again with minor differences between Cd4- and Cd8a-defined subsets of selection intermediates (Fig. 3b, right). Together, these data indicate active TCR signalling in both $Cd4^+ Cd8a^-$ and $Cd4^- Cd8a^+$ selection intermediates.

Expression of the Il7r gene was significantly elevated in $Cd4^+ Cd8a^+$, $Cd4^+ Cd8a^-$ and $Cd4^- Cd8a^+$ selection intermediates compared to pre-selection DP (Supplementary Fig. 5a). Conversely, expression of Socs1, the gene encoding the suppressor of cytokine signalling SOCS1, was significantly reduced in $Cd4^+ Cd8a^-$ and $Cd4^- Cd8a^+$ selection intermediates compared to pre-selection DP (Supplementary Fig. 5a). The expression of Gimap genes, which are targets of IL7R signalling, was also significantly elevated in $Cd4^+ Cd8a^+$, $Cd4^+ Cd8a^-$ and $Cd4^- Cd8a^+$ selection intermediates compared to pre-selection DP (Supplementary Fig. 5a). Members of the JAK and STAT families and a range of cytokines showed evidence for activation (Supplementary Fig. 5b), and so did TGFβ/SMAD and IFNγ, which are among the cytokines that can support the differentiation and/or survival of CD8 lineage thymocytes[20] (Supplementary Fig. 5b). Significant activation scores downstream of STATs and IFNγ in $Cd4^- Cd8a^+$ selection intermediates and of JAKs, TGFβ/SMAD and other cytokines in both $Cd4^+ Cd8a^-$ and $Cd4^- Cd8a^+$ subsets suggest that cytokine signalling pathway activity is

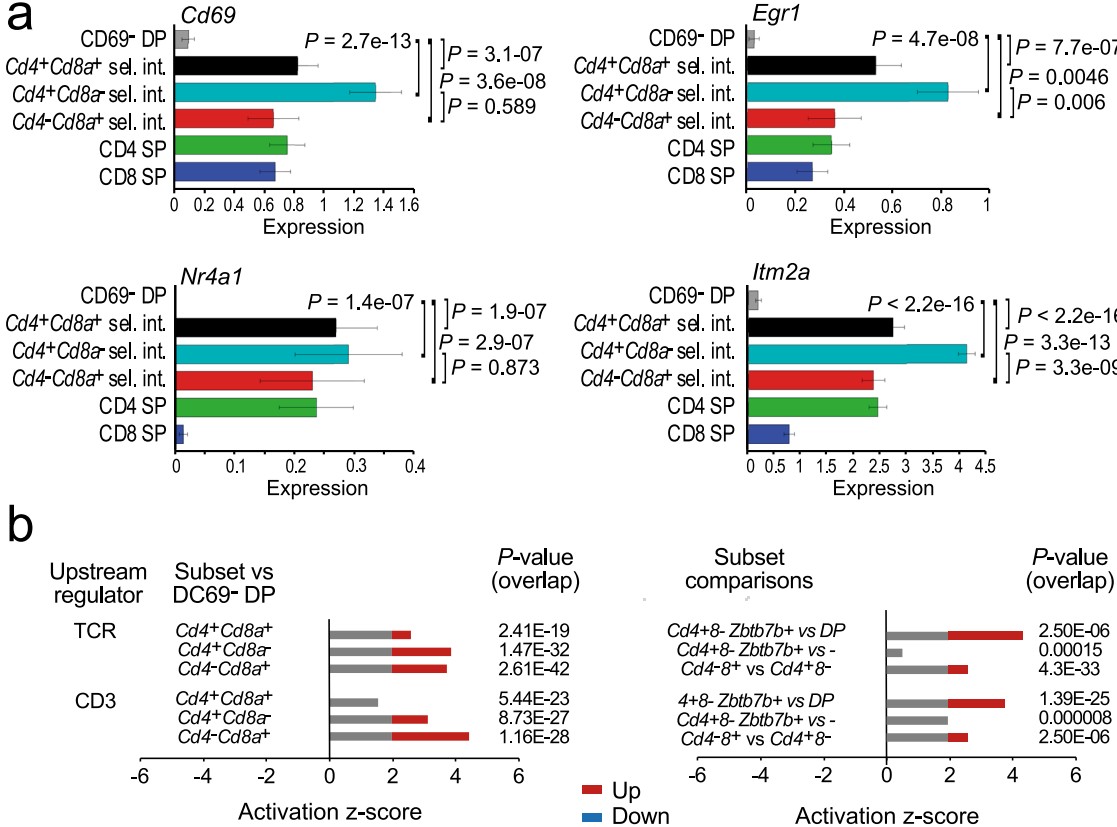

**Fig. 3 Expression of TCR signalling genes by selection intermediates of defined coreceptor gene expression status. a** Expression of TCR activation genes Cd69, Egr1, Nr4a1 and Itm2a by selection intermediates of the indicated coreceptor status. Means and standard errors are shown. P-values are derived by two-sided Wilcoxon rank-sum test. Cell numbers are listed in Supplementary Tables 2 and 3. Source data are provided as a Source Data file. **b** IPA analysis of pathway activity downstream of the TCR and of CD3. Activation z-scores above 2 and below −2 are considered significant. Selection intermediates versus CD69− DP is shown on the left. Where available, comparisons between subsets of selection intermediates are shown on the right. Source data are provided as a Source Data file.

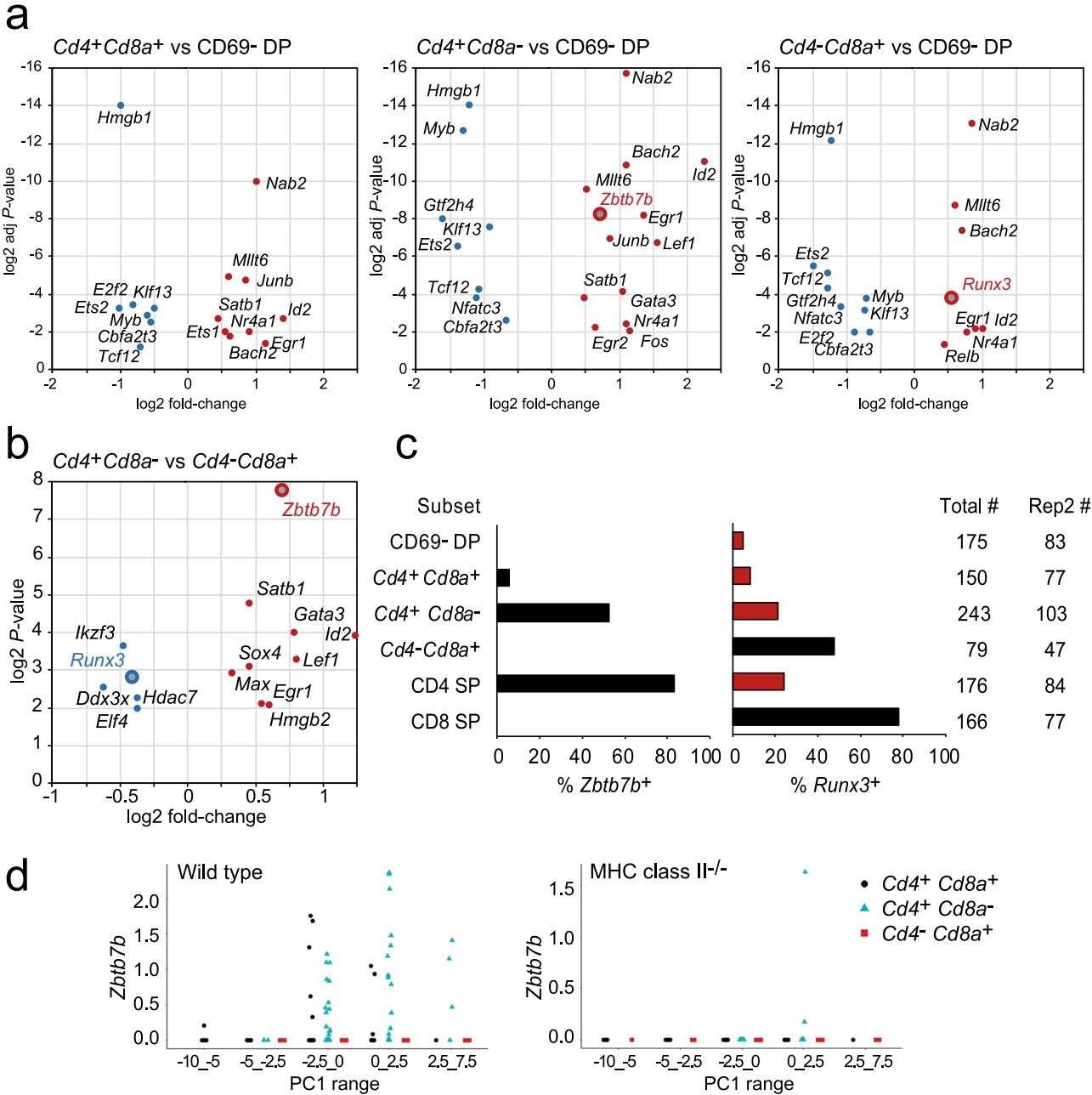

**Fig. 4 Lineage-specific expression of *Zbtb7b* but not *Runx3*. a** Differential expression of transcriptional regulators in *Cd4+Cd8a+* (left), *Cd4+Cd8a−* (middle) and *Cd4−Cd8a+* selection intermediates (right) compared to CD69− DP. Shown are log2 fold-changes and adjusted *P*-values (two-sided Wilcoxon rank-sum test). *Zbtb7b* and *Runx3* are highlighted. Source data are provided as a Source Data file. **b** Differential expression of transcriptional regulators in *Cd4+Cd8a−* versus *Cd4−Cd8a+* selection intermediates. Shown are log2 fold-changes and nominal *P*-values (two-sided Wilcoxon rank-sum test). *Zbtb7b* and *Runx3* are highlighted. Source data are provided as a Source Data file. **c** The frequency of *Zbtb7b* (left) and *Runx3* expression (right) in the indicated thymocyte subsets is shown for wild-type replicate 2. Black: 'lineage-appropriate' expression. Red: 'lineage-inappropriate' expression. Source data are provided as a Source Data file. **d** The expression of *Zbtb7b* by selection intermediates of the indicated coreceptor gene expression status for different PC1 ranges in wild type (left) and MHC class II−/− thymus (right). Source data are provided as a Source Data file.

not restricted to *Cd4− Cd8a+* selection intermediates, and therefore not strictly linked to coreceptor reversal[21–23].

**Lineage-specific expression of *Zbtb7b* but not *Runx3*.** The transcription factors *Zbtb7b* and *Runx3* are central to CD4 and CD8 lineage specification, respectively[24–27]. In wild-type selection intermediates, *Zbtb7b* was significantly upregulated in the *Cd4+ Cd8a−* subset, while *Runx3* was significantly upregulated in the *Cd4− Cd8a+* subset of selection intermediates (Fig. 4a). Accordingly, *Zbtb7b* and *Runx3* were differentially expressed

between *Cd4+ Cd8a−* versus *Cd4− Cd8a+* selection intermediates (Fig. 5b). *Zbtb7b* was expressed in *Cd4+ Cd8a+* and *Cd4+ Cd8a−* selection intermediates and CD4 SP thymocytes (Fig. 4c, d), but was completely absent from pre-selection DP, *Cd4− Cd8a+* selection intermediates, and CD8 SP thymocytes (Fig. 4c). *Zbtb7b* was almost completely absent from selection intermediates in MHC class II-deficient thymi (Fig. 4d), indicating that MHC class II was required for the expression of *Zbtb7b* by selection intermediates[28]. Hence, *Zbtb7b* expression was restricted to the CD4 lineage, and its induction required MHC class II, consistent with

its role as a CD4 lineage-specifying transcription factor[24,25,28]. In contrast, *Runx3* transcripts were detected not only in CD8 lineage cells, but also in CD4 SP, *Cd4+ Cd8a+* and *Cd4+ Cd8a−* selection intermediates (Fig. 4c). These *Runx3* transcripts originated predominantly from the distal *Runx3* promoter. *Runx3* expression in CD4 lineage cells (CD4 SP and *Cd4+ Cd8a−* selection intermediates) was significantly more frequent (63/419) than *Zbtb7b*

expression in CD8 lineage cells (0/245 CD8 SP and *Cd4− Cd8a+* selection intermediates, $P = 4.58e{-}14$, two-sided Fisher's Exact test for count data). According to kinetic signalling models, the expression of *Runx3* is linked to coreceptor reversal from *Cd4+ Cd8a−* to *Cd4− Cd8a+* and to CD8 lineage differentiation[4]. The presence of *Runx3* in *Cd4+ Cd8a+* and *Cd4+ Cd8a−* selection intermediates and CD4 SP thymocytes indicates that *Runx3*

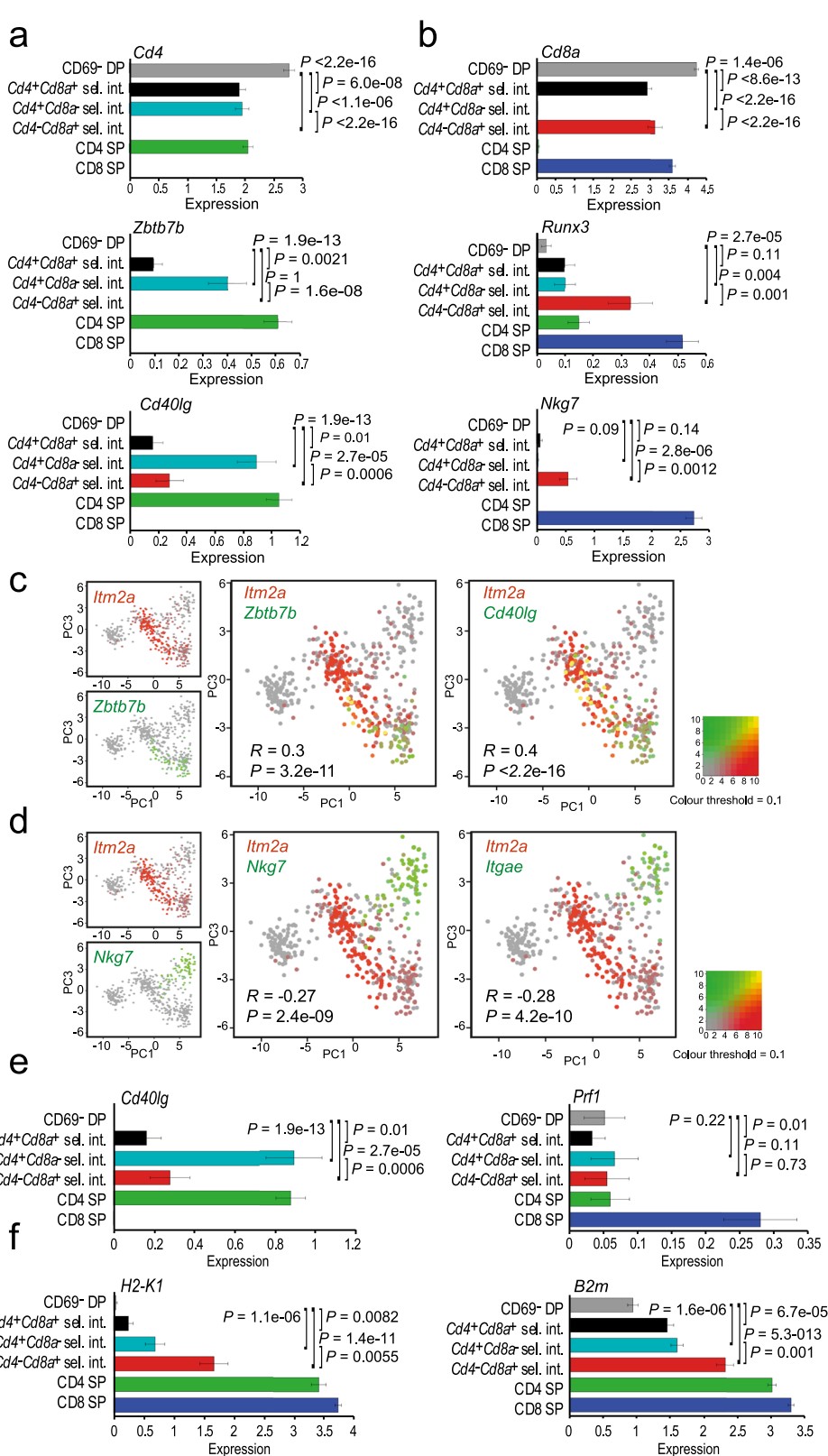

**Fig. 5 CD4 and CD8 lineage characteristics are established at different times during CD4/CD8 lineage choice and differentiation. a** Expression of *Cd4*, *Zbtb7b* and *Cd40lg* by selection intermediates of the indicated coreceptor status. Means and standard errors are shown. *P*-values are derived by two-sided Wilcoxon rank-sum test. Cell numbers are listed in Supplementary Tables 2 and 3. Source data are provided as a Source Data file. **b** Expression of *Cd8a*, *Runx3* and *Nkg7* by selection intermediates of the indicated coreceptor status. Means and standard errors are shown. *P*-values are derived by two-sided Wilcoxon rank-sum test. Cell numbers are listed in Supplementary Tables 2 and 3. Source data are provided as a Source Data file. **c** Co-expression of the activation marker *Itm2a* with the CD4 lineage marker *Zbtb7b* and of *Cd40lg* ($R = 0.30$, $P = 3.2e{-}11$ and $R = 0.40$, $P < 2.2e{-}16$, respectively). See Supplementary Fig. 6a for a depiction of the positive correlation between *Itm2a* and the CD4 lineage markers *Zbtb7b* and *Cd40lg*. Source data are provided as a Source Data file. **d** Lack of co-expression of the activation marker *Itm2a* with the CD8 lineage markers *Nkg7* and *Itgae* ($R = -0.27$, $P = 2.4e{-}09$ and $R = -0.28$, $P = 4.2e{-}10$, respectively). See Supplementary Fig. 6b for a depiction of the negative correlation between *Itm2a* and the CD8 lineage markers *Nkg7* and *Itgae*. Source data are provided as a Source Data file. **e** Mean expression of *Cd40lg*, involved in T cell help, and the cytotoxic T cell marker *Prf1* in selection intermediates with distinct coreceptor gene expression. Means and standard errors are shown. *P*-values are derived by two-sided Wilcoxon rank-sum test. Cell numbers are listed in Supplementary Tables 2 and 3. Source data are provided as a Source Data file. **f** Mean expression of *H2-K1* and *B2m* in selection intermediates with distinct coreceptor gene expression. Means and standard errors are shown. *P*-values are derived by two-sided Wilcoxon rank-sum test. Cell numbers are listed in Supplementary Tables 2 and 3. Source data are provided as a Source Data file.

expression is not strictly lineage-specific, and that the induction of *Runx3* is not linked to coreceptor reversal.

**CD4 and CD8 lineage characteristics are established at different times during CD4/CD8 lineage choice and differentiation.** Gene expression in $Cd4^+ Cd8a^-$ and $Cd4^- Cd8a^+$ subsets of selection intermediates showed a modest ($R = 0.17$) but highly significant correlation with CD4 SP and CD8 SP populations, respectively (Supplementary Table 4) with overall coherence in the direction of regulation ($P < 2.2e{-}16$, Asymptotic Linear-by-Linear Association Test). $Cd4^+ Cd8a^-$ selection intermediates showed higher expression of the CD4 lineage marker *Zbtb7b* and of *Cd40lg* than $Cd4^- Cd8a^+$ selection intermediates (Fig. 5a). By contrast, $Cd4^- Cd8a^+$ selection intermediates showed higher expression of *Runx3* and the CD8 lineage marker *Nkg7* (Fig. 5b). These data indicate that $Cd4^+ Cd8a^-$ selection intermediates have features of CD4 lineage cells, and $Cd4^- Cd8a^+$ selection intermediates have features of CD8 lineage cells.

To address the temporal relationship between thymocyte activation and lineage specification, we examined the dynamics of CD4 and CD8 lineage marker gene expression at the single-cell level. We found that the CD4 lineage marker *Zbtb7* and also *Cd40lg* were frequently expressed alongside activation markers, as illustrated by *Itm2a* (Fig. 5c). The expression of *Zbtb7b* and *Cd40lg* was positively correlated with *Itm2a* (Fig. 5c and Supplementary Fig. 6a). In contrast, CD8 lineage markers such as *Nkg7* and *Itgae* were upregulated largely after the expression of *Itm2a* had subsided, and the expression of *Nkg7* and *Itgae* was negatively correlated with *Itm2a* (Fig. 5d and Supplementary Fig. 6b). *Cd40lg* (CD154) is a mediator of T cell help[29], and acquired by $Cd4^+ Cd8a^-$ selection intermediates (Fig. 5e). Granzyme B (*Gzmb*) and perforin (*Prf1*) mediate cytotoxic T cell functions. While *Gzmb* expression is acquired by CD8 T cells post-thymically, *Prf1* expression was largely restricted to CD8 SP (Fig. 5e). The same was seen for *Itgae* (CD103), which forms a heterodimer that interacts with E-cadherin and facilitates cytotoxic functions[30,31]. Hence, the CD4 lineage characteristics examined here were acquired concomitantly with activation during CD4/CD8 lineage choice and differentiation, whereas CD8 lineage characteristics were acquired largely subsequent to activation. To ask how patterns of coreceptor expression relate to maturation state, we compared the expression of *H2-K1* and *B2m* between $Cd4^+ Cd8a^-$ and $Cd4^- Cd8a^+$ selection intermediates. *H2-K1* and *B2m* were more highly expressed in $Cd4^- Cd8a^+$ than in $Cd4^+ Cd8a^+$ or $Cd4^+ Cd8a^-$ selection intermediates (Fig. 5f), suggesting that $Cd4^- Cd8a^+$ selection intermediates are more mature than the $Cd4^+ Cd8a^+$ or $Cd4^+ Cd8a^-$ subsets.

**Waves of coreceptor gene expression during CD4/CD8 lineage choice and differentiation.** To address the order and the timing of coreceptor gene activity patterns during CD4/CD8 lineage choice and differentiation, we projected the expression of *Cd4* and *Cd8a* onto the principal components PC1 versus PC2 (Fig. 6a), and PC1 versus PC3 (Fig. 6b). Visual inspection suggested that $Cd4^+ Cd8a^-$ cells appear before $Cd4^- Cd8a^+$ cells along the maturation axis PC1. This was supported by average Euclidian distances between subsets of selection intermediates and SP thymocytes (PC1 and PC3: $Cd4^+ Cd8a^-$ selection intermediates: 6.08 to CD4 SP, 6.93 to CD8 SP; $Cd4^- Cd8a^+$ selection intermediates: 5.43 to CD4 and CD8 SP). To formally evaluate the order of $Cd4^+Cd8a^+$, $Cd4^+Cd8a^-$ and $Cd4^-Cd8a^+$ selection intermediates, we plotted their frequencies and compared their position along the PC1 trajectory with that of pre-selection DP (CD69$^-$ DP), CD4 SP and CD8 SP thymocytes (Fig. 6c). We found that early peaks of $Cd4^+ Cd8a^+$ and $Cd4^+ Cd8a^-$ selection intermediates were followed by a later peak of $Cd4^- Cd8a^+$ selection intermediates (Fig. 6d, $P = 1.35e{-}14$, one-sided Kolmogorov–Smirnov test, Supplementary Fig. 7).

As a complementary approach to gauge the order of coreceptor activity patterns, we examined trajectories identified by the Slingshot algorithm[32] (Fig. 6e, left). Quantification of selection intermediates along pseudotime (Fig. 6e, middle) supported the conclusion that $Cd4^+Cd8a^-$ selection intermediates precede $Cd4^-Cd8a^+$ selection intermediates (Fig. 6e, right, $P = 1.18e{-}08$, one-sided Kolmogorov–Smirnov test).

If this order reflects an inherent program of coreceptor gene expression in TCR-signalled thymocytes with fixed timing, the same sequence of events would ensue even in a setting where all TCR signals are triggered by engagement of MHC class I. Selection intermediates in MHC class II-deficient thymi provide a simple test for this prediction. We therefore repeated the analysis described above with thymocytes from MHC class II-deficient mice. Unexpectedly, we found that $Cd4^+Cd8^-$ and $Cd4^-Cd8^+$ selection intermediates arose simultaneously along PC1 when TCR ligand availability was restricted to MHC class I (Fig. 6f, $P = 0.79$, one-sided Kolmogorov–Smirnov test). This conclusion was corroborated by trajectory analysis (Fig. 6g, $P = 0.77$, one-sided Kolmogorov–Smirnov test). These data suggest that the CD4 and CD8 lineages arise sequentially, but that the timing of this sequence is not hardwired.

**MHC class discrimination by selection intermediates.** Given that the timing of coreceptor gene expression patterns was determined by the availability of MHC class II, we next analysed the expression of TCR activation genes as a proxy for TCR signalling in wild-type and MHC class II-deficient selection intermediates. MHC class II$^{-/-}$ $Cd4^+Cd8a^+$ selection intermediates

showed significantly lower expression of *Cd69, Cd5* and other TCR activation genes than wild-type *Cd4⁺Cd8a⁺* selection intermediates (Fig. 7a and Supplementary Fig. 8). The expression of the inducible transcription factors *Nr4a1, Egr1* and *Nab2* (a repressor of EGR transcription factors) and the inducible signalling component *Mapk11* was also reduced in MHC class II$^{-/-}$ *Cd4⁺Cd8a⁺* selection intermediates (Fig. 7a). This indicates that *Cd4⁺Cd8a⁺* selection intermediates are capable of MHC class discrimination. Reduced expression of TCR-induced genes was also found in *Cd4⁺Cd8a⁺* selection intermediates that were phenotypically CD4⁺CD8⁺ (Fig. 7b). This indicates that MHC class discrimination by selection intermediates was not

contingent on the loss of either CD4 or CD8 coreceptors. MHC class-dependent differences in TCR activation gene expression in *Cd4⁺Cd8a⁺* selection intermediates were followed by altered transcription factor and cytokine signalling gene expression at the *Cd4⁺Cd8a⁻* stage. MHC class II$^{-/-}$ *Cd4⁺Cd8a⁻* selection intermediates showed increased expression of the IL7R downstream gene *Gimap3*, increased expression of *Runx1*, and a trend towards increased expression of *Runx3* (Fig. 7c). *Runx* transcription factors are known regulators of *Cd4* and *Cd8a* expression[26,27]. Taken together with the finding that *Cd4⁺Cd8⁻* and *Cd4⁻Cd8⁺* selection intermediates appear simultaneously in the MHC class II$^{-/-}$ thymus, these data indicate that the strength

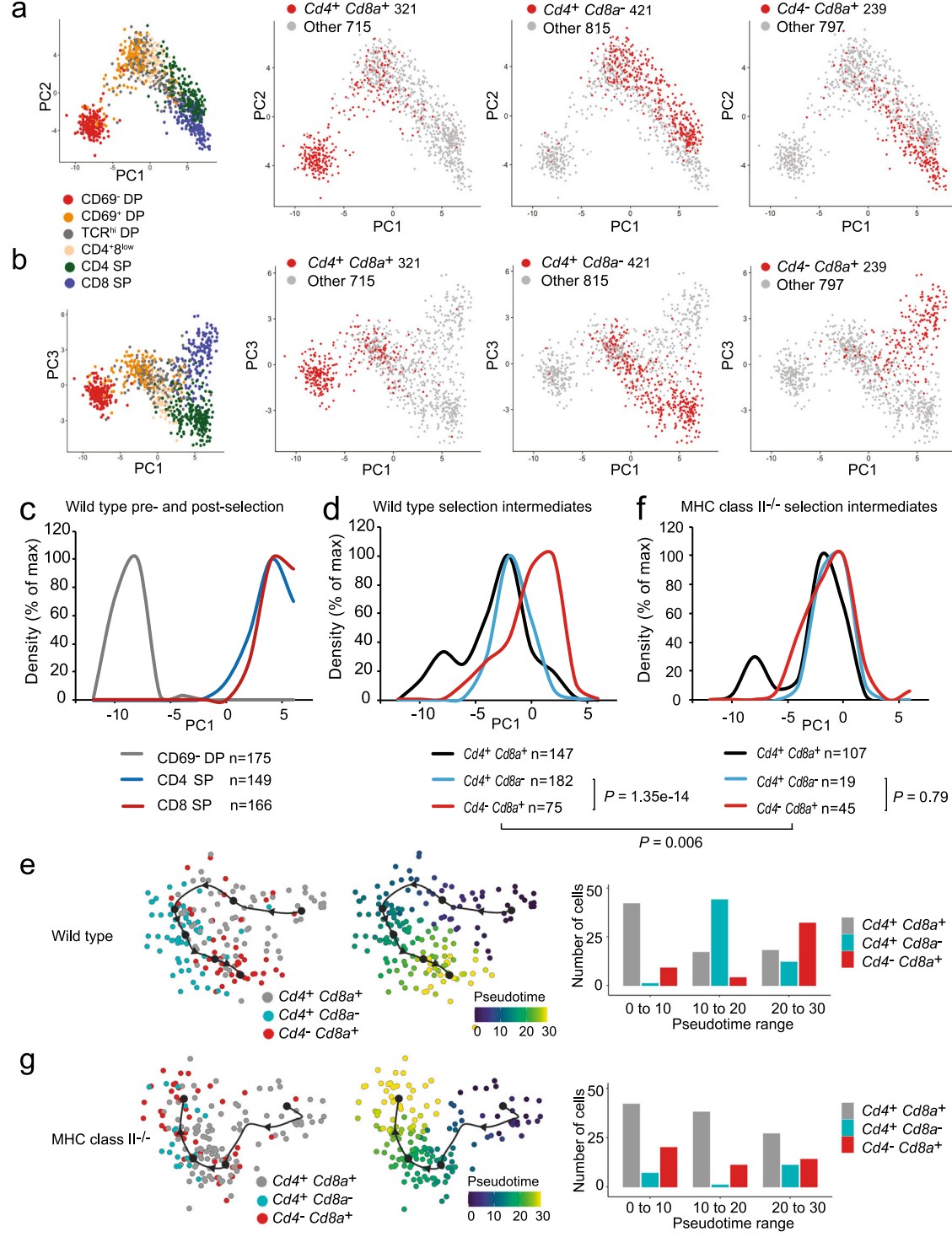

**Fig. 6 The temporal sequence of coreceptor gene expression by selection intermediates. a** PCA 1 versus 2. The inset (far left) shows all cells coloured by cell surface phenotype. Red dots show the position of cells that are $Cd4^+Cd8a^+$ (left), $Cd4^+Cd8a^-$ (centre) or $Cd4^-Cd8a^+$ (right). Source data are provided as a Source Data file. **b** PCA 1 versus 3. The inset (far left) shows all cells coloured by cell surface phenotype. Red dots show the position of cells that are $Cd4^+Cd8a^+$ (left), $Cd4^+Cd8a^-$ (centre) or $Cd4^-Cd8a^+$ (right). Source data are provided as a Source Data file. **c** The temporal sequence of pre- and post-selection wild-type thymocytes. The vertical axis shows the number of pre-selection DP, CD4 SP and CD8 SP expressed normalised to the maximal number of cells detected for each population. The number of cells in each cell population is indicated. Source data are provided as a Source Data file. **d** The temporal sequence of coreceptor gene expression by selection intermediates. The vertical axis shows the number of selection intermediates with the indicated coreceptor gene expression normalised to the maximal number of cells for each gene expression pattern. The number of selection intermediates with each coreceptor gene expression pattern is indicated. *P*-values: one-sided Kolmogorov–Smirnov test. See Supplementary Fig. 7 for individual biological replicates. Source data are provided as a Source Data file. **e** Slingshot trajectory of wild-type $Cd4^+Cd8a^+$, $Cd4^+Cd8a^-$, $Cd4^-Cd8a^+$ selection intermediates based on PCA clustering (top), pseudotime analysis (middle) and quantification of coreceptor gene expression patterns along the pseudotime axis (bottom). *P*-values: one-sided Kolmogorov–Smirnov test. The alternative Slingshot clustering options, MDS and t-SNE, gave equivalent results. **f** The order of coreceptor gene expression patterns is not invariant. The position of MHC class II$^{-/-}$ selection intermediates $Cd4^+Cd8a^+$, $Cd4^+Cd8a^-$, $Cd4^-Cd8a^+$ along PC1. Note the difference between wild-type (**d**) and MHC class II$^{-/-}$ (**f**). *P*-values: one-sided Kolmogorov–Smirnov test. Source data are provided as a Source Data file. **g** Slingshot trajectory of MHC class II$^{-/-}$ $Cd4^+Cd8a^+$, $Cd4^+Cd8a^-$, $Cd4^-Cd8a^+$ selection intermediates based on PCA clustering (top), pseudotime analysis (middle) and quantification of coreceptor gene expression patterns along the pseudotime axis (bottom). *P*-values: one-sided Kolmogorov–Smirnov test. The alternative Slingshot clustering options, MDS and t-SNE, gave equivalent results.

of TCR signals affects the time spent at the $Cd4^+Cd8^-$ selection intermediate stage. Convergence of signal strength and the timing of coreceptor gene expression in selection intermediates unifies key aspects of quantitative and kinetic signalling models[4,13] (summarised in Fig. 7d discussed below).

**Signal strength has the potential to subvert CD4/CD8 lineage choice.** Current kinetic signalling models posit that CD4/CD8 lineage choice is determined exclusively by signal continuity in the face of sequential expression of CD4 and CD8 coreceptors, and not by MHC class discrimination based on signal strength[4,6,7,15,16]. This view was supported by re-engineering of $Cd8a$ to encode the stronger signalling cytoplasmic tail of CD4 (CD8.4; ref. [16]). Greater signal strength increased the number of preselection thymocytes recruited into the selection process and the efficiency of positive selection, but not CD4/CD8 lineage choice in MHC class II-deficient thymocytes or in mice transgenic for the MHC class I-restricted F5 TCR (refs. [16,33]). To further explore whether signal strength can affect lineage choice, we examined the fate of thymocytes we opted for the MHC class I-restricted OT-I TCR because this TCR drives a prominent subset of CD4$^+$CD8$^{low}$ selection intermediates in the presence of wild-type CD8, and our previous studies had shown that CD8.4 enhances TCR-proximal signalling in OT-I transgenic thymocytes while retaining the requirement for MHC class I in positive selection[33]. We examined the generation of CD4 lineage cells in CD8.4 $Rag2^{-/-}$OT-I TCR transgenic mice and found substantial numbers of CD4 SP thymocytes (Supplementary Fig. 9a) and CD4 lymph node T cells (Supplementary Fig. 9b), indicating that signal strength can undermine lineage choice.

## Discussion

A major obstacle to understanding CD4/CD8 lineage choice and differentiation has been that the expression of $Cd4$ and $Cd8$ coreceptor genes is not directly visible, and that the cell surface phenotype of selection intermediates does not reliably indicate coreceptor gene expression[34]. scRNA-seq identified $Cd4$ and $Cd8a$ coreceptor gene transcripts in the great majority of pre- and post-selection thymocytes, indicating very low dropout rates for the coreceptor genes $Cd4$ and $Cd8a$ in our scRNA-seq. The detection of coreceptor transcripts was therefore a strong indicator for the activity of coreceptor genes in individual cells, and allowed the classification of selection intermediates based on coreceptor gene expression. When combined with the position of individual selection intermediates within the framework of

maturation, activation and lineage specification, this provided a direct view of the temporal order of coreceptor gene activity and the associated gene expression programs.

In the unperturbed thymus, $Cd4^-Cd8^+$ selection intermediates appeared significantly later than $Cd4^+Cd8a^-$ selection intermediates. Interestingly, the order and the timing of coreceptor gene expression by selection intermediates was not hardwired, as $Cd4^+Cd8^-$ and $Cd4^-Cd8^+$ subsets arose simultaneously when TCR ligand availability was restricted to MHC class I.

The expression of $Zbtb7b$ was initiated relatively early in $Cd4^+Cd8a^-$ selection intermediates, at a time when selection intermediates still showed abundant expression of activation markers. $Zbtb7b$ expression was absolutely dependent on MHC class II and highly CD4 lineage-specific. In contrast, $Runx3$ was expressed in a substantial fraction of non-CD8 lineage cells. This indicates that $Runx3$ is not strictly a CD8 lineage marker, and that $Runx3$ expression is not directly linked to coreceptor reversal.

$Cd4^+Cd8^-$ and $Cd4^-Cd8^+$ selection intermediates expressed TCR signalling-induced activation markers at significantly higher levels than pre-selection DP thymocytes. This suggests that TCR signalling continues in $Cd4^-Cd8^+$ selection intermediates, even though cytokine signalling is necessary for CD8 lineage differentiation and/or survival[20]. These data disagree with predictions by kinetic signalling models that termination of TCR signalling is essential for CD8 lineage commitment[20] and instead support the view that continued TCR engagement and downstream signalling contribute to CD8 lineage differentiation[35–37].

Early acquisition of CD4 lineage markers and mediators of helper T cell function contrasted with late acquisition of CD8 lineage markers and mediators of CTL function. We did not find co-expression of markers for both the CD4 and the CD8 cell lineage by selection intermediates other than $Cd4$ and $Cd8a$. An interesting interpretation is that CD4 and CD8 lineage programs do not interfere with each other because they are implemented at different times during differentiation.

The expression of TCR activation genes was reduced when class II was absent, even in $Cd4^+Cd8a^+$ selection intermediates that still expressed both CD4 and CD8 on the cell surface. Hence, selection intermediates discriminate between MHC class I and -II, and this discrimination does not require the loss of coreceptor proteins or RNA. The absence of MHC class II also changed the timing of coreceptor gene expression by selection intermediates. We suggest a model that links signal strength and the timing of coreceptor gene expression in selection intermediates. In this scenario, the strength of TCR signals affects the time spent at the $Cd4^+Cd8^-$ selection intermediate stage (Fig. 7d). This model

links MHC class discrimination to coreceptor gene expression where $Cd4^+Cd8a^+$ selection intermediates can distinguish between MHC classes before—and perhaps without—downregulation of $Cd8$ expression. Cytokine signals are implicated in the expression of $Runx3$, which acts to repress $Cd4$ and to activate $Cd8a$ (refs. [26],[27]). MHC class II$^{-/-}$ selection intermediates showed a notable trend for increased expression of the IL7R downstream gene $Gimap3$ as well as $Runx1$ and $Runx3$ as

potential mediators of coreceptor switching from $Cd4^+Cd8a^-$ to $Cd4^-Cd8a^+$ (refs. [4],[20],[22]).

The ability of thymocytes to discriminate between MHC classes based on signal strength was a cornerstone of instructive models for CD4/CD8 lineage choice[1], and an important feature of proposals that lineage choice is based on signal strength and duration[9–12]. Lineage choice by signal strength subsequently fell out of favour, largely because experiments with chimeric coreceptors

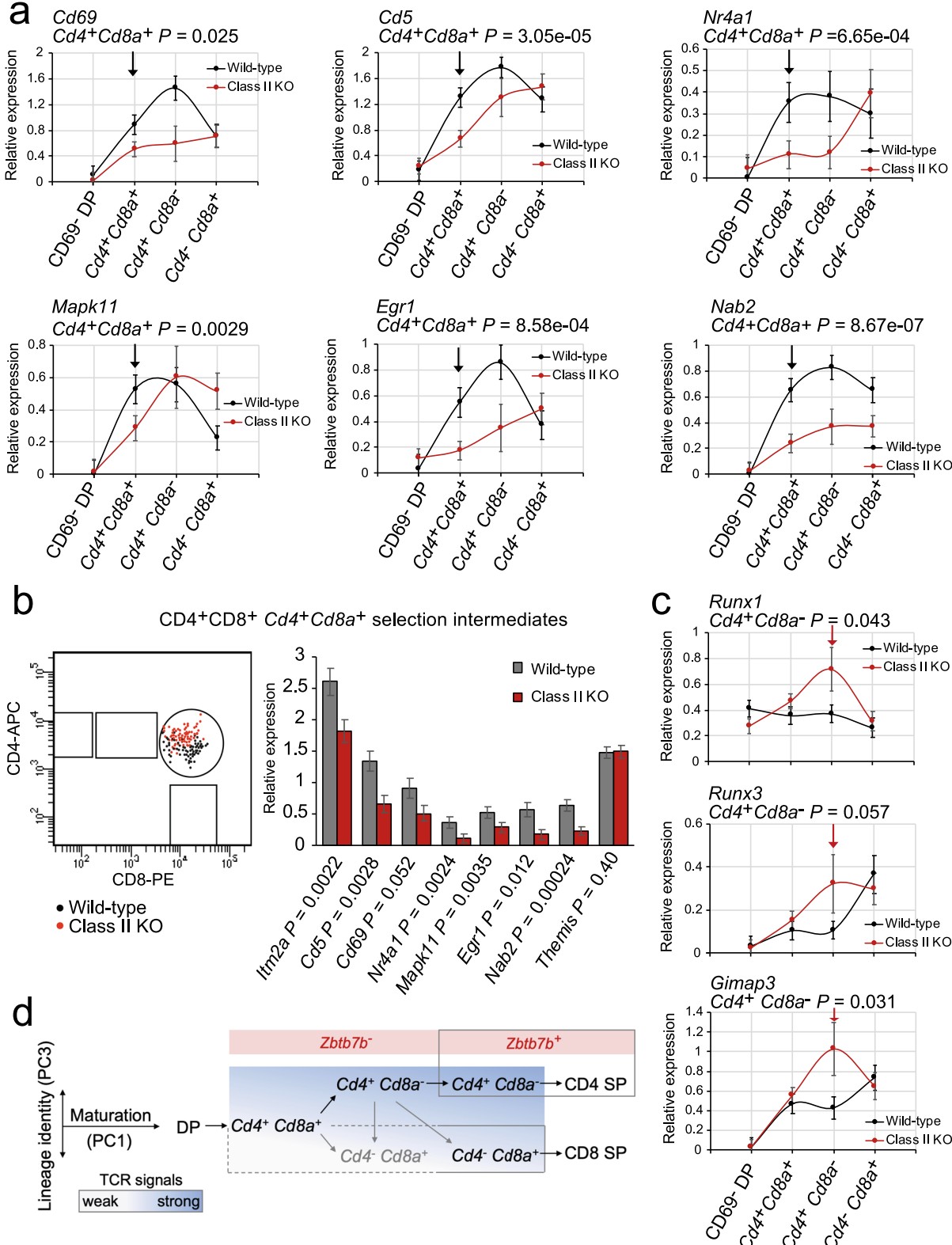

**Fig. 7 MHC class discrimination and *Cd4 Cd8a* coreceptor gene expression in selection intermediates. a** Expression of TCR activation genes in CD69⁻ DP thymocytes and selection intermediates of the indicated coreceptor status in wild-type (black) and MHC class II-deficient thymi (red). Means and standard errors are shown. *P*-values (one-sided Wilcoxon rank-sum test) are for $Cd4^+Cd8a^+$ selection intermediates. Cell numbers are listed in Supplementary Tables 2 and 3. Source data are provided as a Source Data file. **b** Expression of cell surface CD4 and CD8 (left) and activation markers, transcription factors and signalling components (right) by CD69⁺DP and TCR^hi DP selection intermediates that retain expression of both *Cd4* and *Cd8a* coreceptors at the RNA level (wild-type $n = 73$, MHC class II-deficient $n = 77$). Themis is shown to illustrate that not all selection-related genes showed differential expression between MHC class II-deficient and wild-type selection intermediates. Means and standard errors are shown. *P*-values: one-sided Wilcoxon rank-sum test. Source data are provided as a Source Data file. **c** Expression of *Runx1, Runx3* and *Gimap3* in CD69⁻ DP thymocytes and selection intermediates of the indicated coreceptor status in wild-type (black) and MHC class II-deficient thymi (red). Means and standard errors are shown. *P*-values (one-sided Wilcoxon rank-sum test) are for $Cd4^+Cd8a^-$ selection intermediates. Cell numbers are listed in Supplementary Tables 2 and 3. Source data are provided as a Source Data file. **d** A working model that combines key aspects of signal strength and sequential coreceptor models by proposing that the signal strength is linked to the dynamics of coreceptor gene expression during CD4/CD8 lineage choice and differentiation. Boxed areas indicate dependence on MHC class II. The emergence of $Zbtb7b^+Cd4^+Cd8a^-$ selection intermediates requires MHC class II (top). The strength of TCR signals affects the time spent at the $Cd4^+Cd8^-$ selection intermediate stage (darker blue indicates stronger signals, bottom). Grey arrows indicate possible transitions between coreceptor activity states that cannot be inferred directly from the data.

suggested that increased signal strength failed to subvert CD4/CD8 lineage choice[4,16]. Our own data show that the chimeric CD8.4 coreceptor diverts thymocytes expressing the MHC class I-restricted OT-I TCR differentiate to the CD4 lineage. We suggest that weaker TCR signals downstream of MHC class I engagement trigger accelerated transition to $Cd4^-Cd8a^+$ coreceptor status and in this way minimise the risk of inappropriate lineage choice.

In summary, our single cell view of CD4/CD8 lineage choice and differentiation shows that selection intermediates discriminate MHC classes prior to the termination of *Cd4* or *CD8a* expression and adjust the timing of coreceptor gene expression accordingly. These findings bring together key aspects of quantitative signalling models where differences between TCR/CD4/MHC class II signals and TCR/CD8/MHC class I signals determine lineage choice and differentiation of DP thymocytes[13,38–40] and of kinetic signalling models where lineage choice is aided by a program of sequential expression of coreceptors[4]. Signal strength alone does not fully explain lineage choice[4,6,7,9] but contributes to correct lineage outcome, as indicated by mismatches between TCR specificity and lineage choice in settings where signal strength is perturbed (e.g. ref. [15], this study). We suggest that such mismatches may be minimised by the strategy uncovered here, which is to link transitions between coreceptor gene activity to signal strength.

## Methods

**Flow cytometry and cell sorting**. Thymocyte cell suspensions were stained for 20 min at room temperature with CD4-APC, CD8a-PE, TCRb-BV421 and CD69-FITC (BD-Pharmingen). Single cells were sorted into 96-well plates containing lysis buffer using a FACSAria Fusion flow cytometer (BD Biosciences) and the gates depicted in Supplementary Fig. 1. Flow cytometry standard files were analysed with DIVA (BD Biosciences) and FlowJo v10 (TreeStar Inc) analysis software.

C57BL/6 (C57BL/6OlaHsd, Envigo, UK) and Mice lacking MHC class II expression[41] (JAX stock #003584) were maintained separately under specific pathogen-free conditions under Project Licences issued by the Home Office, UK. OT-I $Rag2^{-/-}$ and CD8.4 OT-I $Rag2^{-/-}$ mice[32] were maintained together at the Institute of Molecular Genetics of the Czech Academy of Sciences, Prague, in accordance with laws of the Czech Republic. Six-week-old male or female mice were killed by cervical dislocation, and thymocyte or lymph-node cell suspensions were stained with CD4-APC, TCRb-FITC, CD69-BV421 (Pharmingen) or CD4-Alexa Fluor 700, CD8a-PE or CD8a-BV421, TCRb-APC (Biolegend), and LIVE/DEAD NIR (ThermoFisher). Data acquisition was on a Cytek Aurora flow cytometer (Cytek Biosciences) and flow cytometry standard files were analysed with FlowJo v10 (TreeStar Inc) analysis software. Data were further analysed using GraphPad Prism v5.04 (GraphPad Software).

**scRNA-seq libraries and sequencing**. Wild-type replicate 1: Full-length single-cell RNA-seq libraries were prepared using the Smart-seq2 protocol[42] with minor modifications. Briefly, freshly harvested single cells were sorted into 96-well plates containing the lysis buffer (0.2% Triton-100, 1 U/µl RNase inhibitor). Reverse transcription was performed using SuperScript II (ThermoFisher Scientific) in the presence of 1 µM oligo-dT30VN (IDT), 1 µM template-switching oligonucleotides (QIAGEN) and 1 M betaine. cDNA was amplified using the KAPA Hifi Hotstart ReadyMix (Kapa Biosystems) and IS PCR primer (IDT), with 25 cycles of amplification. Following purification with Agencourt Ampure XP beads (Beckmann Coulter), product size distribution and quantity were assessed on a Bioanalyzer using a High Sensitivity DNA Kit (Agilent Technologies). A total of 140 pg of the amplified cDNA was fragmented using Nextera XT (Illumina) and amplified with Nextera XT indexes (Illumina). Products of each well of the 96-well plate were pooled and purified twice with Agencourt Ampure XP beads (Beckmann Coulter). Final libraries were quantified and checked for fragment size distribution using a Bioanalyzer High Sensitivity DNA Kit (Agilent Technologies). Pooled sequencing of Nextera libraries was carried out using a HiSeq2000 (Illumina) to an average sequencing depth of 0.5 million reads per cell. Sequencing was carried out as paired-end (PE75) reads with library indexes corresponding to cell barcodes.

Wild-type replicate 2 and MHC class II⁻/⁻: Full-length single-cell RNA-seq libraries were prepared using the SMART-Seq v5 Ultra Low Input RNA (SMARTer) Kit for Sequencing (Takara Bio). All reactions were downscaled to one quarter of the original protocol and performed following thermal cycling manufacturer's conditions. Cells were sorted into 96-well plates containing 2.5 µl of the Reaction buffer (1× Lysis Buffer, RNase Inhibitor 1 U/µl). Reverse transcription was performed using 2.5 µl of the RT MasterMix (SMART-Seq v5 Ultra Low Input RNA Kit for Sequencing, Takara Bio). cDNA was amplified using 8 µl of the PCR MasterMix (SMART-Seq v5 Ultra Low Input RNA Kit for Sequencing, Takara Bio) with 25 cycles of amplification. Following purification with Agencourt Ampure XP beads (Beckmann Coulter), product size distribution and quantity were assessed on a Bioanalyzer using a High Sensitivity DNA Kit (Agilent Technologies). A total of 140 pg of the amplified cDNA was fragmented using Nextera XT (Illumina) and amplified with double indexed Nextera PCR primers (IDT). Products of each well of the 96-well plate were pooled and purified twice with Agencourt Ampure XP beads (Beckmann Coulter). Final libraries were quantified and checked for fragment size distribution using a Bioanalyzer High Sensitivity DNA Kit (Agilent Technologies). Pooled sequencing of Nextera libraries was carried out using a HiSeq4000 (Illumina) to an average sequencing depth of 0.5 million reads per cell. Sequencing was carried out as paired-end (PE75) reads with library indexes corresponding to cell barcodes.

**Data analysis**. Smart-seq2 and SMARTer sequencing data were aligned with TopHat2 version 2.1.1 (ref. [43]) that uses the bowtie2 version 2.3.4.3 for alignment[44]. We used the GRCm38 (mm10) mouse genome reference for alignment and gene annotation from UCSC. Read counts for genes were calculated using velocyto version 0.17 (ref. [45]), for SMARTer sequencing data cells with total read counts <500,000 or >1500,000 were excluded, and the cell-gene count matrix obtained was used for downstream analysis.

We performed Seurat v3 (ref. [46]) pre-processing (log-normalisation using variance stabilising transformation method), and identified highly variable (F-set) genes individually for each wild-type replicate (Supplementary Data 1). The highly variable genes from replicate 2 were selected and read counts from both replicates were piped to Seurat v3 standard integration workflow in order to generate integrated PCA for replicate 1 and replicate 2. We chose F-set from replicate 2 because replicate 2 scRNA-seq data were generated using the more sensitive SMARTer protocol compared to replicate 1 WT Smartseq2 data.

Using read counts for F-set genes, we applied Seurat v3 standard integration workflow for integrated analysis of replicate 1 WT Smartseq2 and replicate 2 WT/MHC class II⁻/⁻ SMARTer datasets. The pipeline identified shared cell states that were present between replicate 1 and 2 by generating a batch-corrected expression matrix for F-set genes in all cells and enabling them to be jointly analysed. We then scaled the integrated data, ran PCA and visualised the results.

To find differentially expressed genes between sorted thymocyte subsets and between selection intermediates classified by coreceptor status, we applied Seurat v3 FindMarkers function on replicate 2 WT using two-sided Wilcoxon rank-sum test without cut-off threshold on log-fold-change. Heat maps were generated with Seurat v3 DoHeatmap function using log-normalisation values for candidate genes. FeaturePlot function with blend option was applied for visualisation of co-expression of gene pairs.

To find genes that changed along PC1, PC2 and PC3, we tested the null hypothesis of no change along PCs for each gene using the gam R package. Top 1000 genes by *P*-value were selected to create contingency tables with activation, differentiation and lineage-specific gene markers for Fisher Exact test.

Slingshot (ref. [32]) was used for trajectory inference and pseudotime analysis of selection intermediates, based on a recent benchmark study of 57 trajectory inference methods[47].

**Reporting summary**. Further information on research design is available in the Nature Research Reporting Summary linked to this article.

## Data availability
scRNA-seq data have been deposited at GEO under accession number GSE149207. Population RNA-seq data has been deposited at GEO under accession number GSE154670. All other data are included in the supplemental information or available from the authors upon reasonable request. Source data are provided with this paper.

## Code availability
Custom Rmd scripts used for generating figures are available from Github: https://github.com/LMSBioinformatics/ScRNAseq-SMARTer-Analysis.

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

## Acknowledgements

We thank Dr. B. Seddon (University College London) for discussion, Dr. J. Merkenschlager (Rockefeller University) for critical reading of the manuscript, the referees for valuable suggestions, and Dr. G. Kassiotis (Crick Institute, London) for MHC class II$^{-/-}$ mice. Supported by the Medical Research Council, UK (B.L., A.G.F., M.M.), the Wellcome Trust (Investigator Award 099276/Z/12/Z to M.M.), the National Institutes of Health (NIH 1R35GM136284 to M.K.), and the Swiss National Science Foundation (SNSF Promys IZ11Z0_166538 to O.S.).

## Author contributions

Y.G., V.H., S.R.G., B.P., J.E., and M.M. carried out the experiments; M.M.K., X.C., H.A.P., V.H., Y.-F.W., G.R.-E., I.R.-R., R.G., M.H.D., D.D., M.S.K., O.S., and M.M. analysed and curated the data; M.M.K., V.H., G.R.E., I.R.-R., D.D., M.S.K., O.S. and M.M. visualised the data; M.M.K., L.B., M.S.K., B.L., H.H., A.G.F., O.S. and M.M. conceptualised the paper; M.M.K., V.H., O.S. and M.M. wrote the paper.

## Competing interests

The authors declare no competing interests.
