## [Peer Review File · Nature Communications]

REVIEWERS' COMMENTS

Reviewer #1 (Remarks to the Author):

This is a revised manuscript by Karimi et al utilizing scRNA-Seq approaches to explore lineage commitment in CD4 versus CD8+ thymocytes. Overall, I think this is a much improved manuscript. I am not confident that they have entirely proved their model but the new data does provide a provocative new hypothesis that is worth further exploration by the field. The authors have dealt with most of the concerns raised in the initial reviews. There are a few issues and some minor corrections that I feel the authors should deal with.

1. Regarding original point 3. My initial concern was that - The data seem insufficient to completely back up the claims. The authors suggest a sequential model and argue against a single branchpoint. An alternative model could be that loss of CD8 expression in the CD4+CD8- intermediates is the test for lineage commitment cells with MHC-II reactive TCRs can continue to receive TCR stimulation and commit towards the CD4 lineage while those that fail to receive that signal upregulate the CD8 lineage program. This would be in line with previous reports suggesting that loss of TCR signal and adoption of the CD8 program are linked to cytokine responsiveness.

The authors responded:

Our new data show that selection intermediates discriminate MHC class prior to the loss of coreceptor expression. They suggest a model where TCR signal strength informs the timing of coreceptor gene expression and CD4/CD8 lineage choice.

Our response to revision - I like the idea proposed here but am not totally convinced. Perhaps showing this data by looking at individual cells would be more valuable, i.e. if some cells look "normal" for TCR signaling in the early intermediates in the MHC-II KO this would support that some cells, presumably MHC-I reactive, can receive a signal while others cannot. This would help the argument that cells are "discriminating" MHC reactivity in these early stages. Such data would help shore up their model if available.

2. With regard to point 4, the authors have added new data showing that IL7R-responsive transcripts are equivalently induced in both signaling intermediates. This is a substantial improvement. I still think it would be nice to add protein data regarding IL7R if possible.

3. Regarding minor point 3, Line 143- "Fig. S3c, we note that the Runx3 transcripts detected originated to > 99% from the distal Runx3 promoter with the exception of a single CD8 SP cell that contained 29.2% pRunx3)." It is unclear what point is being made here but this data could just be directly added to supplemental figure 3.

The authors responded: We have simplified the description of Runx3 transcripts in the revised manuscript.

This is still discussed in the text with no explanation of why this is noteworthy. While it is perhaps not a big problem it is unclear what it adds. Moreover, the data related to this statement is not being shown. Without the data or relevance this statement should probably be eliminated entirely.

New comment:

1- "By contrast, Cd4- Cd8a+ selection intermediates showed higher expression of Runx3 and the CD8 lineage marker Nkg7 (Fig. 4b)."- Is this supposed to reference figure 5b?

Reviewer #2 (Remarks to the Author):

In the revised manuscript by Karimi et al, authors significantly improved data analyses. In particular they expanded analyses to further characterize Cd4+Cd8- and Cd4-Cd8+ selection intermediates emerged in Wt and MHC class II KO mice and added results of OT-I CD8.4 mice. Having these new data set, authors explained well what is significance of their focused single cell RNA-seq (scRNA-seq). I also appreciate improvement of discussion section. I am glad to recommend this manuscript for publication in Nature Communications after minor textual revision.

Minor points:

1. There are mistakes of figure labelling in the text. : Line 354-355, I think they should be Fig 5a and Fig 5b.; Line 363 Fig. S5a should be S6a.;Line 425, Fig 7c should be 6c.; Line 426 add Fig. S7.; Line 430, this should be Fig. 6e left.
2. It is better to show Cd40lg in Fig. 4a.
3. In line 513 and 633,Runx1 is described as a downstream molecules of IL7R signaling. Please add reference supporting this. Increase of Runx1 in Cd4+Cd8a- intermediated in MHC-II KO than in Wt mice might stem from increased frequency of NKT cell in Cd4+Cd8a- intermediated in MHC-II KO mice. Please clarify this point by examining NKT markers such as Plzf expression.
4. In paragraph from line 604 -610, authors might be interested in discussing recent paper (Kojo et al. Life Sci Alliance. (2020) 3:e202000642) that described effect of constitutive CD8ab expression on thymocyte differentiation.

Reviewer #3 (Remarks to the Author):

I have read with interest the content of this manuscript and the authors' answers to the previous referees. I find the investigation well thought-out and of excellent technical quality. The manuscript is very well written, though I would encourage the author to be more explicit in certain conclusions and definitions and use a more accessible, less restricted-to-a-few-language when talking about the underlying concepts/principle on mechanisms proposed to explain thymocytes development.

The work intelligently exploits exhaustive and rigorously a plethora of transcriptome data by RNA sequencing of FACS single cell-sorted thymocyte subsets to try and extract most useful information on their lineage fate decisions. The authors made the clever technical choice of relying on a reasonable sampling size to obtain good quality data, rather than huge data set with potential of losing important information. The work makes use of rigorous statistical tests, taking into account only robustly significant statistical data.

The authors have re-visited the complex process of CD4/CD8 DP thymocytes' fate dilemma, upon TCR engagement with self-pMHC. This event informs about the gene expression program driving DP thymocytes towards becoming mature CD4 or CD8 (SP) thymocytes, the precursors of the two main branches of T-cell adaptive immunity. Merckenschlager and co-workers provide a first example of how to use most advanced single-thymocyte transcriptome data analysis to trace their developmental process. Once deposited and available, the data will represent an excellent source of information for

many other studies.

Early work demonstrated that initial instruction (as opposed to stochasticity) is provided to DP thymocytes by a "stronger" cell activation signal when TCR engages, together with CD4, with self pMHC class II. This event pushes towards a maturing CD4 thymocyte. However, if this is not the case, (e.g., expression on the DP cell of a TCR more fit to engage self-peptide-class I aided by CD8) there should be a chance for a DP thymocyte to move to an intermediate stage (or selection intermediate). The present work points to a selection intermediate for both lineages (a sort of "stand-by" status) being decisive to then progress to a thymocyte becoming SP. Interestingly and different from previous models, the authors provide indications from transcriptome analysis that when selection intermediates start moving away from DP cells they also start to better discriminate MHC I and II ligands and this (as it may make sense) happens before expression of Cd4 or CD8a is terminated. The timing to achieve a definitive decision would be dictated by coreceptor gene expression. The transcriptome data also exclude a decisive role of certain cytokines present in the thymus as driving CD8 SP choice as previously proposed.

This mechanism can be interpreted as the co-receptors being decisive to increase signal amplitude and duration, by decreasing TCR-pMHC off-rates and at the same time boosting signalling by co-receptor-borne Lck. This dual signaling contribution (duration and amplitude that are likely to reinforce each other) may make a real difference for pushing the choice decision. But this potential mechanism is not discussed. It is unclear if this is a deliberate choice due to conjecture based on their data or because the authors do not fully appreciate the underlying mechanism of co-coreceptors function as positive modulators of TCR signalling. The authors could discuss this by citing published data by Cheng-Zhu group and Gascogne's group suggesting that TCRs signaling is necessary to "attract" co-receptors. As the authors point out, these data may reconcile quantitative signaling models (signal strength) and the kinetics (duration) of signaling events to accomplish DP thymocytes decisive choices before becoming SP as signal strength alone does not fully explain lineage choice, but perhaps modify it to avoid incorrect co-receptor and TCR matching.

I have only a few comments and suggestions for minor changes. However, I strongly recommend publication of this paper without hesitation.

Minor points

- 1 - it would be most appropriate to designate TCR signaling capability as a composite of "signal amplitude and duration", rather than the defining generically as "strong" or "weak". Indeed, it is possible that the magnitude of signal amplitude and duration dictates gene pattern expression.
- 2 - I found the authors quite reticent to explicit fully their thinking behind their main conclusions. I would encourage then to provide not only generic conclusions but to dare adding more about mechanistically and how they see the signalling aspects in the model. I feel that this attitude takes away some "juice" from their excellent contribution to understand the complex process of thymocyte fate choice.
- 3 - I found quite a number of mismatches between figure number and numbers indicated in the text. I suggest the authors to carefully read their text and repair these mistakes.
- 4 - page 5, 156 : "...differentiation and quantifies the relative contribution of each component to the overall variance of gene expression". I suggest that the author refer here to a figure and/or qualify in a short sentence what they really mean.
- 5 - I found at times the authors using "coded/ultra-specialised language" accessible only to a few. It would help to be a bit more explicit while conserving succinct phrasing.
- 6 - it would help to provide a suppl. table listing the "activation", "maturation" and "lineage identity" genes and the genes that define the selection intermediates. Without a clear tangible definition by gene expression it is harder to connect these categories with real defining facts.

7 – when considering why a response of DP cells to class II should be “stronger”, it is surprising that there has been no challenge to the idea of a better co-receptor function based on better Lck interaction while some recent data do not support this view. Other mechanism may be possible and it would be interesting to hear comments on this issue in this paper.

Response to REVIEWERS' COMMENTS

We thank the referees for their patience and for their constructive comments

Reviewer #1 (Remarks to the Author):

This is a revised manuscript by Karimi et al utilizing scRNA-Seq approaches to explore lineage commitment in CD4 versus CD8+ thymocytes. Overall, I think this is a much improved manuscript. I am not confident that they have entirely proved their model but the new data does provide a provocative new hypothesis that is worth further exploration by the field. The authors have dealt with most of the concerns raised in the initial reviews. There are a few issues and some minor corrections that I feel the authors should deal with.

1. Regarding original point 3. My initial concern was that - The data seem insufficient to completely back up the claims. The authors suggest a sequential model and argue against a single branchpoint. An alternative model could be that loss of CD8 expression in the CD4+CD8- intermediates is the test for lineage commitment cells with MHC-II reactive TCRs can continue to receive TCR stimulation and commit towards the CD4 lineage while those that fail to receive that signal upregulate the CD8 lineage program. This would be in line with previous reports suggesting that loss of TCR signal and adoption of the CD8 program are linked to cytokine responsiveness.

The authors responded:

Our new data show that selection intermediates discriminate MHC class prior to the loss of coreceptor expression. They suggest a model where TCR signal strength informs the timing of coreceptor gene expression and CD4/CD8 lineage choice.

Our response to revision - I like the idea proposed here but am not totally convinced. Perhaps showing this data by looking at individual cells would be more valuable, i.e. if some cells look “normal” for TCR signaling in the early intermediates in the MHC-II KO this would support that some cells, presumably MHC-I reactive, can receive a signal while others cannot. This would help the argument that cells are “discriminating” MHC reactivity in these early stages. Such data would help shore up their model if available.

We thank the referee for this suggestion. TCR signals are required for pre-selection DP to enter the selection process and to become selection intermediates. In the MHC class II-deficient thymus, these signals will be through MHC class I. The figure below shows the expression of the activation marker *Itm2a* along PC1, which illustrates the responses by wild-type and MHC class II-deficient *Cd4+ Cd8a+*, *Cd4+Cd8a-* and *Cd4-Cd8a+* selection intermediates. Early *Cd4+ Cd8a+* selection intermediates show reduced expression of *Itm2a* (and other TCR signaling genes, not shown) in the MHC class II-deficient thymus.

Referee Figure. Activation marker expression in individual selection intermediates ordered along the maturation trajectory PC1. Top: The level of mRNA for the activation marker *Itm2a* is shown along the maturation trajectory PC1 for individual selection intermediates classified as Cd4+Cd8a+, Cd4+Cd8a- and Cd4-Cd8a+ in wild-type (left) and MHC class II KO (right) thymus. Bottom: The frequency of mRNA for the activation marker *Itm2a* detected in subsets of selection intermediates classified as Cd4+Cd8a+, Cd4+Cd8a- and Cd4-Cd8a+ in wild-type (left) and MHC class II KO (right) thymus is shown along the maturation trajectory PC1.

2. With regard to point 4, the authors have added new data showing that IL7R-responsive transcripts are equivalently induced in both signaling intermediates. This is a substantial improvement. I still think it would be nice to add protein data regarding IL7R if possible.

We thank the referee for this suggestion. Unfortunately it would require significant further methods development to provide IL7R protein data for selection intermediates classified by coreceptor gene activity.

3. Regarding minor point 3, Line 143- “Fig. S3c, we note that the Runx3 transcripts detected originated to > 99% from the distal Runx3 promoter with the exception of a single CD8 SP cell that contained 29.2% pRunx3.” It is unclear what point is being made here but this data could just be directly added to supplemental figure 3.

The authors responded: We have simplified the description of Runx3 transcripts in the revised manuscript.

This is still discussed in the text with no explanation of why this is noteworthy. While it is perhaps not a big problem it is unclear what it adds. Moreover, the data related to this statement is not being shown. Without the data or relevance this statement should probably be eliminated entirely.

The information is included because we have been asked about promoter usage of *Runx3* transcripts by colleagues in the Runx field

New comment:

1- "By contrast, Cd4- Cd8a+ selection intermediates showed higher expression of Runx3 and the CD8 lineage marker Nkg7 (Fig. 4b)."- Is this supposed to reference figure 5b?

We thank the referee for spotting this error and have corrected it in the revised manuscript

Reviewer #2 (Remarks to the Author):

In the revised manuscript by Karimi et al, authors significantly improved data analyses. In particular they expanded analyses to further characterize Cd4+Cd8- and Cd4-Cd8+ selection intermediates emerged in Wt and MHC class II KO mice and added results of OT-I CD8.4 mice. Having these new data set, authors explained well what is significance of their focused single cell RNA-seq (scRNA-seq). I also appreciate improvement of discussion section. I am glad to recommend this manuscript for publication in Nature Communications after minor textual revision.

Minor points:

1. There are mistakes of figure labelling in the text. : Line 354-355, I think they should be Fig 5a and Fig 5b.; Line 363 Fig. Fig. S5a should be S6a.;Line 425, Fig 7c should be 6c.; Line 426 add Fig. S7.; Line 430, this should be Fig. 6e left.

We thank the referee for spotting these errors and have corrected it in the revised manuscript

2. It is better to show Cd40lg in Fig. 4a.

The focus of Fig 4 is on transcription factors and chromatin proteins.

3. In line 513 and 633, Runx1 is described as a downstream molecules of IL7R signaling. Please add reference supporting this.

The sentence concerned reads 'MHC class II^{-/-} Cd4⁺Cd8a⁻ selection intermediates showed increased expression of the IL7R downstream gene *Gimap3*, *Runx1*, and a trend towards increased expression of *Runx3* (Fig. 7c)'. It describes only one gene, *Gimap3*, as a downstream target of the IL7R. However, to prevent any misunderstanding, we have changes this sentence to read 'MHC class II^{-/-} Cd4⁺Cd8a⁻ selection intermediates showed increased expression of the IL7R downstream gene *Gimap3*, increased expression of *Runx1*, and a trend towards increased expression of *Runx3* (Fig. 7c)'

Increase of Runx1 in Cd4+Cd8a- intermediated in MHC-II KO than in Wt mice might stem from increased frequency of NKT cell in Cd4+Cd8a- intermediated in MHC-II KO mice. Please clarify this point by examining NKT markers such as Plzf expression.

As detailed in our previous response to the referees, we examined the expression of the NKT cell markers *Klrb1*, *Klrb1a*, *Klrb1b* or *Klrb1c* in wild-type or MHC class II-deficient thymi. The only subset that contained cells expressing NKT markers were MHC class II-deficient CD4SP, a population which we did not analyse in this study. There were no selection intermediates that expressed or *Klrb1*, *Klrb1a*, *Klrb1b* or *Klrb1c* in wild-type or MHC class II-deficient thymi. NKT cells are therefore not discussed in the revised manuscript.

4. In paragraph from line 604 -610, authors might be interested in discussing recent paper

(Kojo et al. Life Sci Alliance. (2020) 3:e202000642) that described effect of constitutive CD8ab expression on thymocyte differentiation.

We thank the referee for pointing out this very interesting paper. We note that iNKT cell differentiation in response to constitutive coreceptor expression is somewhat remote from the scope of our study.

Reviewer #3 (Remarks to the Author):

I have read with interest the content of this manuscript and the authors' answers to the previous referees. I find the investigation well thought-out and of excellent technical quality. The manuscript is very well written, though I would encourage the author to be more explicit in certain conclusions and definitions and use a more accessible, less restricted-to-a-few-language when talking about the underlying concepts/principle on mechanisms proposed to explain thymocytes development.

The work intelligently exploits exhaustive and rigorously a plethora of transcriptome data by RNA sequencing of FACS single cell-sorted thymocyte subsets to try and extract most useful information on their lineage fate decisions. The authors made the clever technical choice of relying on a reasonable sampling size to obtain good quality data, rather than huge data set with potential of losing important information. The work makes use of rigorous statistical tests, taking into account only robustly significant statistical data.

The authors have re-visited the complex process of CD4/CD8 DP thymocytes' fate dilemma, upon TCR engagement with self-pMHC. This event informs about the gene expression program driving DP thymocytes towards becoming mature CD4 or CD8 (SP) thymocytes, the precursors of the two main branches of T-cell adaptive immunity. Merckenschlager and co-workers provide a first example of how to use most advanced single-thymocyte transcriptome data analysis to trace their developmental process. Once deposited and available, the data will represent an excellent source of information for many other studies.

Early work demonstrated that initial instruction (as opposed to stochasticity) is provided to DP thymocytes by a "stronger" cell activation signal when TCR engages, together with CD4, with self pMHC class II. This event pushes towards a maturing CD4 thymocyte. However, if this is not the case, (e.g., expression on the DP cell of a TCR more fit to engage self-peptide-class I aided by CD8) there should be a chance for a DP thymocyte to move to an intermediate stage (or selection intermediate). The present work points to a selection intermediate for both lineages (a sort of "stand-by" status) being decisive to then progress to a thymocyte becoming SP. Interestingly and different from previous models, the authors provide indications from transcriptome analysis that when selection intermediates start moving away from DP cells they also start to better discriminate MHC I and II ligands and this (as it may make sense) happens before expression of Cd4 or CD8a is terminated. The timing to achieve a definitive decision would be dictated by coreceptor gene expression. The transcriptome data also exclude a decisive role of certain cytokines present in the thymus as driving CD8 SP choice as previously proposed.

This mechanism can be interpreted as the co-receptors being decisive to increase signal amplitude and duration, by decreasing TCR-pMHC off-rates and at the same time boosting signalling by co-receptor-borne Lck. This dual signaling contribution (duration and amplitude that are likely to reinforce each other) may make a real difference for pushing the choice

decision. But this potential mechanism is not discussed. It is unclear if this is a deliberate choice due to conjecture based on their data or because the authors do not fully appreciate the underlying mechanism of co-coreceptors function as positive modulators of TCR signalling. The authors could discuss this by citing published data by Cheng-Zhu group and Gascogne's group suggesting that TCRs signaling is necessary to "attract" co-receptors.

The idea that signal duration and amplitude reinforce each other is fascinating. We chose not to discuss this concept in depth as our study is mainly based on RNA-seq data, which do not allow us to distinguish between signal duration and amplitude.

As the authors point out, these data may reconcile quantitative signaling models (signal strength) and the kinetics (duration) of signaling events to accomplish DP thymocytes decisive choices before becoming SP as signal strength alone does not fully explain lineage choice, but perhaps modify it to avoid incorrect co-receptor and TCR matching. I have only a few comments and suggestions for minor changes. However, I strongly recommend publication of this paper without hesitation.

Minor points

1 - it would be most appropriate to designate TCR signaling capability as a composite of "signal amplitude and duration", rather than the defining generically as "strong" or "weak". Indeed, it is possible that the magnitude of signal amplitude and duration dictates gene pattern expression.

The idea that signal duration and amplitude dictate coreceptor gene expression is fascinating. We chose not to discuss this concept in depth as our study is mainly based on RNA-seq data, which do not allow us to distinguish between signal duration and amplitude.

2 - I found the authors quite reticent to explicit fully their thinking behind their main conclusions. I would encourage them to provide not only generic conclusions but to dare adding more about mechanistically and how they see the signalling aspects in the model. I feel that this attitude takes away some "juice" from their excellent contribution to understand the complex process of thymocyte fate choice.

We thank the referee for this encouraging comment. Our restraint is motivated by the concern that our study is mainly based on RNA-seq data, and that very specific models of signaling events would require complementary approaches.

3 - I found quite a number of mismatches between figure number and numbers indicated in the text. I suggest the authors to carefully read their text and repair these mistakes.

We thank the referee for noting these errors and have corrected them in the revised manuscript

4 - page 5, 156 : "...differentiation and quantifies the relative contribution of each component to the overall variance of gene expression". I suggest that the author refer here to a figure and/or qualify in a short sentence what they really mean.

The paragraph the referee refers to was intended as a summary of Figure 1. To clarify our intent we have removed the gap between the description of Fig. 1 and the summary statement. We have also re-written the sentence quoted by the referee as follows '... our

analysis quantifies the relative contribution of each (maturation, activation and lineage identity) to the overall variance of gene expression, and shows that scRNA-seq data alone are sufficient to derive a coherent framework for CD4/CD8 lineage choice and differentiation'.

5 – I found at times the authors using “coded/ultra-specialised language” accessible only to a few. It would help to be a bit more explicit while conserving succinct phrasing.

We apologise if we have used 'ultra-specialised language' and have carefully edited the narrative to address this.

6 – it would help to provide a suppl. table listing the “activation”, “maturation” and “lineage identity” genes and the genes that define the selection intermediates. Without a clear tangible definition by gene expression it is harder to connect these categories with real defining facts.

We thank the referee for this suggestion. Supplementary Table 5 of the revised manuscript shows genes that are transiently up-or downregulated in selection intermediates (activation) or differentially expressed between CD4 and CD8 SP (lineage identity).

Selection intermediates were identified and isolated based on phenotype (CD69+ DP, CD4+8lo, TCRhi DP) as shown in Supplementary Figure 1. The classification of selection intermediates by coreceptor gene activity was based on the expression of *Cd4* and *Cd8a* transcripts alone. We hope this clarifies these important points.

7 – when considering why a response of DP cells to class II should be “stronger”, it is surprising that there has been no challenge to the idea of a better co-receptor function based on better Lck interaction while some recent data do not support this view. Other mechanism may be possible and it would be interesting to hear comments on this issue in this paper.

We thank the referee for this suggestion. Mechanisms of coreceptor function are the subject of active investigation in the Ondrej Stepanek's lab and will be the focus of a separate manuscript.